# Experimental study on the effect of Ivermectin on cattle dung faunas in Eastern Ethiopia

**Shiret Belete[1], Hunde Adugna[1], Tsedalu Yirsa**[1], *

1 School of Veterinary Medicine, Woldia University, Woldia, Ethiopia

* tsedyirsa@gmail.com

## Abstract

An experimental investigation was conducted from December 2023 to June 2024 at the beef farm of Haramaya University. The bulls were divided into two groups: one group received ivermectin treatment, while the other acted as a control. The SUMIVER brand of ivermectin was administered subcutaneously at 1 ml per 50 kg of body weight. Dung samples from both groups were collected and prepared for analysis at five-day intervals during the eighth sampling period. The results were measured, documented, and analyzed both qualitatively and quantitatively. All data were entered and analyzed using an independent t-test with the STATA-14 software. In the qualitative assessment, no dipteran larvae or pupae were detected in the dung samples, except during the eighth round. However, various species of invertebrates were attracted to the freshly managed dung, resulting in a higher visitation rate in the control group compared to the experimental group. In the initial weeks of the study, the presence of ivermectin residues affected the dung beetle population. Conversely, the treated dung showed a significant infestation of termites. After several weeks, both groups of dung pats were invaded by ants, with the control group experiencing infestation first. These findings suggest that ivermectin residues released into the environment through cattle dung can influence negatively plant germination, dung fauna, and soil fertility. Therefore, it is crucial for veterinarians to be knowledgeable about the environmental side effects of ivermectin and to offer guidance to livestock owners.

## 1 Introduction

### 1.1 Background of Study

Gastrointestinal tract infections remain a significant parasitic threat to grazing ruminants worldwide which are adversely affecting their welfare and overall health [1]. The use of anthelmintic (AH) products for disease prevention is essential in managing the gastrointestinal tract (GIN) in ruminants to prevent both significant, and subclinical illnesses [2]. The two main categories of anthelmintics, benzimidazoles, and macrocyclic lactones, collectively account for a market share of around €470 million [3]. Benzimidazoles represent a wide range of antihelminthic agents, including albendazole (ABZ), ricobendazole, mebendazole, fenbendazole, and thiabendazole [4].

**Data availability statement:** The data collected and utilized to support this study is freely accessible within the paper.

**Competing interests:** All authors have no competing interests.

**Abbreviations:** AH, Antihelmentic; ABZ, Alebendazole; AVM, Avermectin; CYP, Cytochrome P450; EVM, Epromectin; GIN, Gastrointestinal nematode; IVM, Ivermectin

Various widely used anthelmintics, including abamectin, ivermectin (IVM), eprinomectin (EPM), and doramectin are classified as disaccharide derivatives of avermectins, which fall under the category of macrocyclic lactones. These compounds are derived from the fermentation of *Streptomyces avermitilis* [5]. Ivermectin was the first avermectin-based product to be commercially available, having been launched in 1981 [6]. It is employed in the treatment of various infestations, including lymphatic filariasis, strongyloidiasis, trichuriasis, ascariasis, head lice, and scabies [7]. It is also used for the treatment of gastrointestinal roundworms, as well as bovine lung-worms and other ectoparasites such as ticks [8]. In addition to its antiparasitic properties, IVM has shown effectiveness in vitro against several viruses, including Zika, Dengue, Influenza, and HIV [9]. Given the potential impact of parasitic diseases on livestock populations and productivity, there exists a significant financial motivation for farmers to integrate these medications into their management practices [10]. The efficacy, safety profile, and novel mechanism of action of ivermectin have quickly established it as a preferred treatment for controlling nematodes and other arthropods in livestock [8].

Animals eliminate anthelmintics through their faeces and urine, with excretion levels ranging from 60% to 90% of the administered dose [11]. The bioavailability of ivermectin is affected by the formulation and administration route [12]. When administered orally, ivermectin typically results in pronounced peaks in excretion with the majority of the dose being expelled over several days. In addition, about 90% of ivermectin and its metabolites are excreted via feces, while only about 1% is eliminated through urine [13]. For injectable or topical formulations, peak excretion of ivermectin generally occurs between two and seven days after treatment, followed by a prolonged elimination phase that may extend beyond four to six weeks. Conversely, in sustained-release formulations, peak elimination may occur several weeks post-treatment [14]. Bile serves as the primary excretion pathway for ivermectin [15]. The metabolism of protease inhibitors, including ivermectin-22, is facilitated by cytochrome P450 (CYP) 3A enzymes, which also metabolize various other medications [16].

The discharge of manure and other untreated diffuse sources frequently contributes to the release of veterinary medications into the environment [17]. The application of ivermectin (IVM) in agricultural practices has been associated with ecotoxicological effects on non-target organisms, particularly affecting insects and bee populations [14]. Ivermectin enters the environment primarily through feces and manure [18]. Its low solubility in water and strong affinity for organic matter in soil lead to its accumulation [15]. The environment's degradation of ivermectin is affected by aerobic and anaerobic conditions, soil-feces mixtures, and exposure to sunlight which promote photodegradation [18].

Dung represents a metabolic byproduct that encompasses a diverse array of microorganisms and their metabolic byproducts, endogenous debris, both partially digested and undigested food particles [19]. The characteristics of dung, including its consistency, size, shape, quantity, chemical composition, and moisture content, exhibit considerable variability. The decomposition of dung is influenced by abiotic factors including location, seasonal variations, soil type, and prevailing meteorological conditions, including airflow, temperature, and precipitation [20]. Many species, including insects, nematodes, earthworms, mites, bacteria, fungi, and various arthropods like spiders and millipedes utilize animal faeces [18].

Insects represent the most adept group in utilizing animal dung. The majority of insects that frequent dung are classified within the orders Coleoptera and Diptera, while Hymenoptera and Isoptera are present to a lesser degree [21]. A significant part of the invertebrate community associated with dung consists of dung beetles, which rely on dung as a nutritional source for both their larvae and adult forms [22]. The impact of avermectins on insects ranges from sub-lethal to lethal effects. The duration of these effects and the vulnerability of different species to these agents can vary significantly [23].

Ivermectin (IVM) is often given for conditions that are provisionally diagnosed, frequently without definitive confirmation of the illness. Additionally, in developing countries, its use has occurred without sufficient consideration for the impact on non-target insect populations after its release into the environment. Notably, clinicians commonly prescribe IVM for a wide range of ailments during various veterinary clinic activities. Pet owners frequently ask for IVM for apparently healthy animals, especially for purposes related to fattening. Furthermore, there hasn't been adequate guidance on the potential side effects of IVM on non-target fauna, flora, and their ecological surroundings. These practices have led to the current experimental study, which aims to evaluate the adverse effects of IVM on the environment, specifically focusing on dung pat colonizer fauna, soil fertility, and pasture health. Therefore, the study's objective was to evaluate how ivermectin affects cattle dung fauna and soil fertility by influencing their role in the decomposition process.

## 2 Materials and methods

### 2.1 Study area

The study was conducted at the Beef farm of Haramaya University, situated in the Haramaya district of the Eastern Hararghe Zone in the Oromia Region of Eastern Ethiopia. This district is positioned 14 kilometres west of Harar city and 508 kilometres east of Addis Ababa, with geographical coordinates of 9°24′N 42°01′E (or 9.400°N 42.017°E) and an elevation of 2047 meters above sea level. The district operates under a mixed production system [24]. It was also undertaken from December 2023 to June 2024, located five kilometres from Haramaya town. The area is characterized by abundant vegetation. It was selected due to its high insect population, which facilitates dung colonization. Additionally, the site is enclosed by a cement concrete fence, constructed around 2007 EC, protecting against the intrusion of domestic and wild animals. Before the experiment, the site was cleared of debris and prepared in two parallel series, with a distance of 2 meters between the series and 1 meter between individual sites. The sites were meticulously arranged to ensure proper observation of dung fauna activities while preventing access by birds and other predators.

### 2.2 Study populations

The animal population involved in this study comprised cattle raised for beef at Haramaya University's Beef Farm. From this population, three bulls were designated as the control group, while three others were assigned to the experimental group. All six bulls were zebu cattle characterized by short horns. The age and weight of the bulls were recorded, with weights varying between 260 kg and 340 kg as illustrated in Table 1. The bulls selected for both the experimental and control groups were sourced from block three of the farm, where they had resided for over three months without any prior treatment. This indicates that they had not received any anthelmintic medications during that period. Furthermore, animals that

Table 1. Characteristics of the study cattle at the study area [25].

| Bull ID | Group | Age (months) | Weight (kg) | Ivermectin dose (ml) |
|---------|-------|--------------|-------------|----------------------|
| Bull 1 | Control | 24 | 260 | 5.2 |
| Bull 2 | Control | 26 | 300 | 6 |
| Bull 3 | Control | 30 | 340 | 6.8 |
| Bull 4 | Experimental | 26 | 310 | 6.2 |
| Bull 5 | Experimental | 30 | 340 | 6.8 |
| Bull 6 | Experimental | 24 | 260 | 5.2 |

had been treated with anthelmintic drugs were isolated from the others until the appropriate withdrawal period for the injected medications had elapsed, under the farm's protocols. This practice ensures that the selected bulls remain undisturbed until the experimental samples are required, thereby minimizing variability and bias in the results. The bulls were provided with a diet of straw, concentrated feed, and water, as per standard feeding practices.

## 2.3 Study design and sampling methodology

The experimental study design was implemented to evaluate the effect of ivermectin on dung fauna and soil fertility. It inhabits dung treated with ivermectin residues, in comparison to untreated dung. For methodological consistency, three bulls were designated as the control group, while the remaining three served as the experimental group. The SUMIVER brand of ivermectin (0.2 mg/kg) was administered subcutaneously following the calculation of the appropriate dosage for each bull based on their weight [16]. The decomposition rate of dung was assessed by systematically collecting and analyzing dung pats from two groups of bulls. Fresh dung of 1 kg samples were collected in the morning and transferred to a designated decomposition site. The samples were placed separately for each group at five-day intervals to ensure continuous monitoring of decomposition progression. The experimental period spanned 40 days, consisting of eight sampling intervals (Day 0, 5, 10, 15, 20, 25, 30, and 40). Each dung pat was standardized by weight at the time of collection to ensure uniformity across samples. Initial and final dry weight (g) of each dung pat were measured using a precision balance (Sartorius Entris 2202-1S, ±0.01 g accuracy). Dung samples were inspected for invertebrate colonization (e.g., dung beetles, fly larvae) using a stereo microscope. Finally, Mass loss percentage was calculated as: Mass loss (%) = (Initial dry weight − Remaining dry weight) × 100 [26].

Observation and Quantification of Invertebrate Activity: Dung pats were examined for invertebrate colonization, noting the presence of dung beetles, fly larvae, termites and other macro invertebrates. Each of the individual invertebrates were recorded at each interval.

## 2.4 Data collection of experimental designs

The initial two experimental locations, designated for the controlled and experimental groups respectively, were irrigated one day before the first application (January 10, 2024) to facilitate insect visitation and infestation, given that the experiment was conducted during the dry season. The irrigation continued until the completion of the pat collection, maintaining the same frequency. The pat was gathered in the morning before farm-workers removed it from both groups of bulls five days post-treatment (January 11, 2024), based on the understanding that ivermectin excretion peaks between two and seven days following subcutaneous injection of 0.2 mg/kg [27]. To prevent cross-contamination of samples, four pats were collected from each group of bulls using distinct materials. These four pats were intended to weigh a total of 1 kg. Before placement on the irrigated site, the four pats were homogenized with a small amount of water to enhance uniformity and create a mushy consistency. The experimental section was initially prepared for insect colonization. Both sections were labeled "Ci" for the control group and "UCi" for the treatment group (with "i" representing the series numbers of pats ranging from 1 to 8) as illustrated in Fig 1A and B. The remaining pats were processed in the same manner, following the schedule of January 11, 16, 21, 26, 31, February 5, 10, and 15, while monitoring for invertebrates.

As soon as a new dung pat was formed, the activities of various dung faunas that frequented the pat were observed and recorded. After four days, the dung samples were analyzed for physical alterations, including the development of fractures and cracks. The dung was then turned over to examine the underside for the presence of holes and tunnels created by beetles,

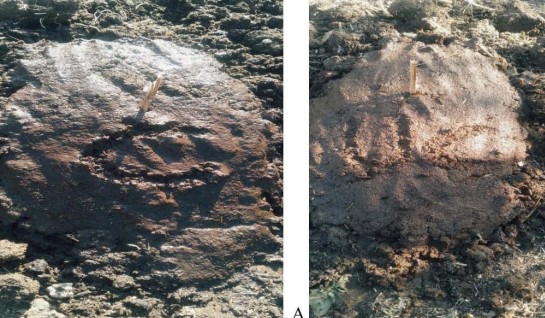

**Fig 1.  Controlled (A) and uncontrolled (B) dung pat made on the first day of the trial.**

termites, and other organisms. The dung was carefully segmented and inspected for insects, larvae, pupae, and newly emerged adults. A small portion was utilized to assess the weight of the dung from two groups by submerging it in water simultaneously, based on the principle that heavier materials sink while lighter ones float [28].

Simultaneously, the release of soil and dung particles was noted in the water bath. The dung samples used for examining the presence of larvae, pupae, and other organisms were fragmented into smaller pieces to facilitate the extraction of larvae and adults from various faunal groups. A small quantity of soil beneath the dung was collected due to the suspicion of pupae migration into it. Hand sorting of adults, larvae, and pupae was performed using sunlight and drops of 10% formalin after the fauna had been extracted, which aided in counting. Additionally, the activities of various flying fauna around the extracted dung were observed for both groups, along with the scent of the extracted dung. Following these procedures, the fields were assessed for water content and the germination of different seeds in areas where dung had been applied. The presence or absence of decomposition of the extracted dung throughout the experimental period was also monitored. The results were documented, recorded, and summarized based on the dung fauna collected during field days, and were analyzed both qualitatively and quantitatively.

## 2.5  Data analysis

All information collected from the designated data collection site was meticulously documented at the specified time intervals. The process of data collection and documentation was executed, followed by inputting the information into Microsoft Office Excel. Subsequently, the analysis was performed using StataMP-14. The data was summarized through experimental data analysis statistics and quantitative data analysis to examine the relationships between variables, employing an independent t-test. This approach was appropriate for this study, which involved two independent variables and multiple outcomes (dependent variables) [29]. A linear regression analysis was performed to assess the relationship between time (days) and mass loss percentage.

## 2.6  Ethics approval and consent to participate

The study, entitled "**Experimental Study on the Effect of Ivermectin on Cattle Dung Faunas in Eastern Ethiopia**", was carried out at the Beef Farm of Haramaya University. The research was conducted by investigators of Shiret Belete, Hunde Adugna, and Tsedalu Yirsa. This study was also conducted following the procedures and animal care standards established by the Federation of Animal Sciences Societies (FASS) [30] guidelines. The study received approval

from the Animal Welfare and Ethical Review Committee of the Department of Veterinary Medicine at Woldia University in Ethiopia **(Ref. No. WDU/CoA/VM/023/05/8/2024)**. All necessary measures were implemented to minimize any pain experienced by the animals involved in the study. Importantly, there were no identified risks or discomforts associated with the sampling process from the study subjects. Furthermore, oral consent was obtained from the participants involved.

## 3  Results

All samples were manually examined for invertebrates, larvae, and pupae using a stereo microscope and physical alterations of samples were observed. Subsequently, factors such as moisture content, weight, cracks, tunnels, holes, and the level of decomposition were evaluated following the extraction of the required fauna. Additionally, the presence or absence of germinated seeds at the dung sites was noted for direct negative effect of ivermectin on dung fuana and indirectly affect seed germination due to killing invertebrates facilitating soil-fertility. No samples were excluded from examination, observation, or measurement.

### 3.1  Qualitative observation of the dung pats

On the initial day of dung production, the dung from the Ivermectin-treated group was processed first, revealing the presence of certain fauna, including Coleoptera and Diptera with reduced stability. Shortly thereafter, dung from the untreated group was produced, which exhibited a significant increase in the activity of Diptera, Coleoptera, and various other faunas. Notably, the fauna that had initially colonized the treated dung returned to the untreated dung, resulting in a diminished movement of dung fauna towards the treated dung. Upon extraction of the dung, larvae and adults were isolated, revealing active flying fauna on the untreated dung, while such activity was markedly lower on the treated dung. The activity of dung fauna persisted until approximately the fifth sample of dung (30 days post-treatment). However, variations were observed among the samples, beginning with the first treated dung sample and continuing with subsequent treated dung samples. This indicates that the activity of dung fauna gradually increased in the treated dung as the duration of post-treatment with Ivermectin extended.

At four-day intervals, the dung was examined, and both types were dried similarly as shown in Fig 2A and B. Advanced cracks developed more prominently in the untreated dung compared to the treated dung. The pattern of fracture formation remained consistent throughout the trials. Upon breaking apart the dung, a shiny, wet, white appearance was noted, accompanied by an ammonia odor from the untreated samples. In contrast, the treated

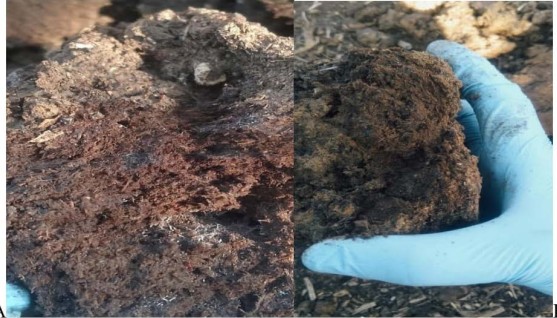

**Fig 2. Formation of white colour of the second round of controlled dung pat after four days (A) and no colour formation of the same round of treated dung pat (B).**

dung exhibited a moist consistency, lacked a white interior, and showed no discernible odor during the initial sampling rounds. After the dung was fragmented into smaller pieces and allowed to settle for a few minutes, the untreated dung dried readily, taking on a brown hue at the surface, whereas the treated dung did not exhibit the same drying characteristics. Observations made a week later revealed that the broken samples of treated dung (from the first, second, and third rounds) showed no signs of decomposition; particularly, the first and second rounds remained unchanged from their initial state on the day of extraction. Conversely, the untreated dung had decomposed significantly. After one week of extraction, both groups were completely dried, and no insect activity was noted on the dried samples.

Upon examination of the larvae, pupae, and adults of various invertebrates, a significant quantity of Diptera larvae was observed, along with different types of adult Coleoptera and varying numbers of fauna in untreated dung, while treated dung showed no such findings until the fourth round of sampling. A few pillbugs, springtails, and slugs were detected in the treated dung starting from the third sample. As the duration of post-treatment increased, the diversity and quantity of fauna observed in the treated dung also rose. Additionally, beetle tunnels were noted within the dung and extending into the ground at the treatment site, although these were fewer compared to untreated dung in the sixth, seventh, and eighth samples. Moreover, the Diptera larvae, which were present in smaller quantities and appeared stunted, were found in the eighth sample of treated dung. These larvae were unable to bury themselves within the dung pieces (inactive) and succumbed shortly after exposure to sunlight. In contrast, a substantial number of termites were identified in the treated dung, along with canals and holes located ventraly in the soil at the site (Fig 3).

No evidence of termites, including their canals and holes, was observed in the untreated dung. Upon examining the extracted dung from the third sample after several weeks, it was noted that the treated dung contained only termite canals, which appeared to be their own, while the untreated dung had newly become infested with termites. As the duration of field activities increased, the presence of termites diminished, with none detected in the treated dung samples from the 6th to the 8th observations. During the ongoing examination of the dung pats, a significant number of ants were noted at the site of the 5th sample of untreated dung, and this presence persisted until the final sample. In contrast, ants were only observed at the sites of the 7th and 8th treated dung pats. There were no significant differences in the characteristics of the ants. The presence of earthworms was minimal in both groups, although the soil beneath the untreated pitfall exhibited a lower number compared to the treated one.

Upon weighing the dung from both groups, a small sample from each was placed in a water bath simultaneously. The dung from the treated group sank rapidly, while the dung from the

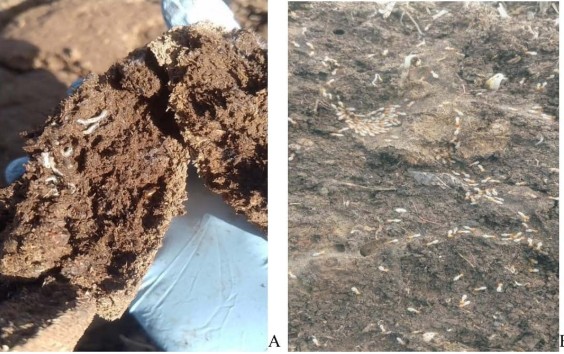

**Fig 3. Diptera larvae of controlled dung pat (A) and huge termite under dung pat of controlled dung pat (B).**

untreated group initially floated on the water's surface during the testing of the first and second samples. As the duration of post-treatment increased, the sinking behaviour of the treated dung diminished, leading to a later tendency to float. During the observation of the sinking and floating dung, the untreated dung released particles and soil quickly, whereas the treated dung did not exhibit the same behaviour, as the untreated dung settled at the bottom of the water. Beneath the untreated dung, well-planted seeds thrived and showed significant growth after a few days. In contrast, the treated dung had fewer planted seeds compared to the untreated group. Nevertheless, as the post-treatment period with ivermectin extended, the presence of planted seeds in the treated group began to correlate with that of the untreated group, illustrated in Fig 4A and B.

## 3.2 Quantitative analysis of the study

All data of the faunas associated with cattle dung were documented numerically. However, this does not imply that every observed and counted fauna was recorded. Insects such as ants, termites, springtails, and lady beetles exhibited high levels of activity, making them challenging to manage and count comprehensively. Various types of dung beetles, exceeding five distinct types, were identified, and their averages were recorded in treatment group (Fig 5) and

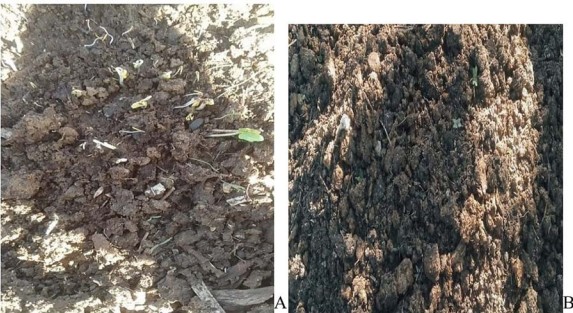

**Fig 4. Germinated seed at the site of controlled dung pat (A) and no germinated seed at the site of treated dung (B).**

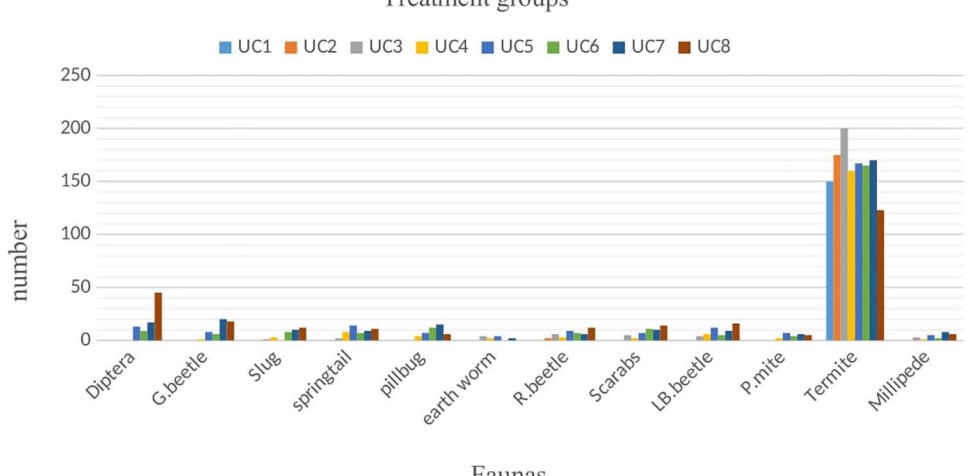

**Fig 5. Graph of treatment group of faunas in cattle dungs in the study area.**

in control group (Fig 6). The data concerning cattle dung faunas collected from both groups were analyzed using StataMP-14 software. The results showed the dung decomposition rates for both the control and ivermectin-treated groups, based on measurements of each dung pat's weight or volume at regular time intervals to assess the effect of ivermectin on dung fauna (Fig 7).

The effect of IVM on dung faunas was analyzed by comparing the dependent variables (dung faunas) of controlled and treated dung pat by using an Independent t-test. All results found were significant (all *P-values* were ≤ 0.05). So, the result found strictly supports the alternative hypothesis and fails the null hypothesis. The results are summarized in Table 2.

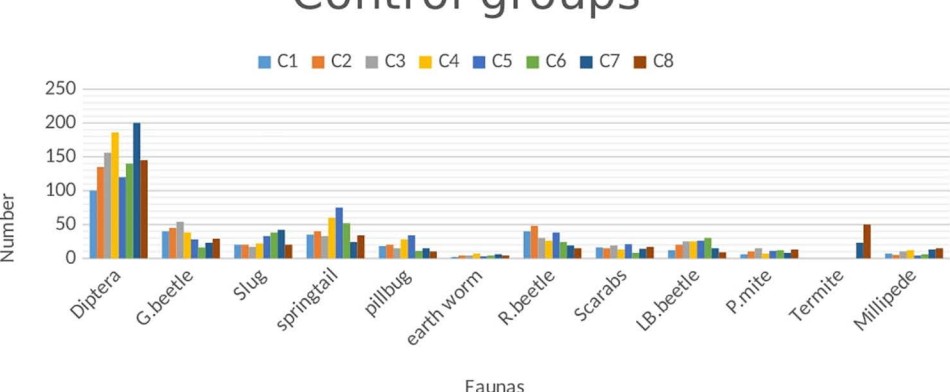

**Fig 6. Graph of controlled group of faunas in cattle dungs in the study area.**

**Fig 7. Dung decomosition rate for control and uncontrolled groups.**

**Table 2. Independent t-test and analysis of variance (ANOVA) results comparing quantitative data between dung faunas from Ivermectin-treated and untreated dung at the study area.**

| Faunas | Mean of control | Mean of experiment | SD of control | SD of experiment | *P-value* |
|---|---|---|---|---|---|
| Dipteral larvae | 30.125 | 0.5 | 5.4876 | 0.707677803 | 0.027 |
| Beetles | 9.375 | 0.5125 | 3.0654 | 0.71589 | 0.001 |
| Pillbug | 4.25 | 1.875 | 2.0688 | 1.369393 | 0.034 |
| Slug | 10.5 | 1.75 | 3.2403 | 1.32226785 | 0.009 |
| Earthworm | 0.5 | 0.375 | 0.70710 | 0.61237435 | 0.009 |
| Millipede | 2.25 | 0.875 | 1.5 | 1.2247448 | 0.013 |
| Predator mite | 3.5 | 1.125 | 1.87082 | 1.0606601 | 0.043 |
| Termite | 0 | 15.5 | 0 | 3.937003 | 0.033 |
| Ants | 8 | 34 | 2.828453 | 5.83097 | 0.048 |

## 4 Discussions

Dung pats were collected from both bulls that had been treated with ivermectin and those that had not in this study. A significant presence of dung fauna was observed on the untreated dung pats, with Diptera being the predominant group, followed closely by other species, aligning with the findings of Saha *et al.* [31]. These invertebrates are attracted to fresh feces as a food source. Additionally, many dung insects are drawn to the fresh dung aroma produced by the microflora in the digestive tract, which may be altered by ivermectin treatment, leading to a decrease in metabolic activity and the production of aromatic compounds. Consequently, the diminished metabolism and reduced scent make it less likely for fresh dung colonizers to inhabit the treated dung. This observation was also concurring with the conclusions drawn by Urrutia *et al.* [32]. Nevertheless, as time elapsed post-treatment, there was an increase in the abundance of fresh dung fauna in the treated dung samples. Previous reports of Holter [33] noted that neither group exhibited fauna when the upper surface dried, as dung fauna relies on the moisture found in fresh dung. Furthermore, research conducted by Beynon *et al.* [34] stated that the dung from ivermectin-treated bulls was not effectively decomposed by detritivores. This suggests that the lack of decomposition in treated dung pats contributes to their high compatibility even after drying, resulting in limited air circulation and organic matter exchange, as well as preventing sunlight from penetrating. These factors contribute to the persistence of the dung's integrity, its muddy consistency, the unpleasant odour, and the failure of seed germination [35].

The persistence of ivermectin implies that residues have the potential to affect the dung community for the entire period. Sutton *et al.* [36] stated that drug residues remained at levels high enough to adversely affect dung-colonizing fauna throughout their entire 180-day trial. He stated that Ivermectin residues had a clear effect on the presence of invertebrates in cowpats. The earlier study reports of Forbes [37] also stated that Diptera larvae were consistently present in higher numbers in cowpats from untreated cattle, and non-dipteran invertebrates such as termites and beetles were generally present in higher numbers in cowpats from treated cattle starting from the first and after times respectively. In addition, Jacobs and Scholtz [38] reported that ivermectin residues eliminate certain dung-dwelling Diptera. These attributed to the low number of Diptera larvae recovered from treated cowpats were a consequence of mortality [34].

Dung beetles represent a highly abundant and specialized segment of the beneficial dung insect community. The previously mentioned studies indicated that no distinct dung beetle species were identified on dung treated with ivermectin, placing them at heightened risk due

to the contamination of dung with ivermectin residues. There was no observed increase in the attraction of dung beetles to feces containing ivermectin; rather, a significant decline in dung beetle biomass was noted throughout the trial period. Additionally, short-term variations were recorded in the total quantity of dung utilized by dung beetles at the designated trial sites. These findings suggest that a single administration of ivermectin to bulls, at the recommended dosage of 0.2 mg/kg of body weight, may lead to a mortality rate among beetles in dung shortly after application, with a reduced mortality rate observed in dung after several weeks. These conclusions are corroborated by different research [39–41]. Nonetheless, the attraction response noted in certain studies appears to be more closely associated with the cellular mode of action of ivermectin.

The potential attraction exhibited by Ivermectin (IVM) may indicate the entrapment of organisms within contaminated faeces. It influences neurotransmitter activity that inhibiting their movement and preventing invertebrates from relocating to uncontaminated areas [42]. According to De Souza and Guimaraes [8] stated that IVM can impair both the locomotor and olfactory functions of the *Scarabaeus cicatricosus* beetle which hinders their ability to carry out essential biological activities. This indicated that climatic conditions significantly impact ecotoxicity assessments. Furthermore, species native to tropical areas often differ from those in temperate zones in terms of reproductive and feeding behaviours, which may lead to varying responses to IVM [43]. Nevertheless, research indicates that organisms from both climatic regions exhibit comparable reactions to IVM [44]. Consequently, it is posited that the attraction or repulsion behaviours observed in dung beetles may stem from an unpredictable side effect or variations in cattle feed [8].

A low population of earthworms was observed in this study, with treated dung not serving as a control for the dung pat. This indicates that earthworms were not significantly present in the treated dung pat and were even less significant in the control group [45]. Upon further examination of other fauna, a substantial number of springtails, millipedes, and pill bugs were recorded during the initial weeks of the trial on untreated dung pat, while fewer were found on the treated dung pat. However, as time passed, no significant differences were noted between the two groups. This observation is consistent with the notion that the impacts of IVMs on springtails (Collembola) and enchytraeids (Anellida) are not as thoroughly documented as those on earthworms; nonetheless, there is some evidence indicating potential toxicity to these organisms [8]. On the other hand, treated dung was significantly infested by termites up to the seventh dung pat. It has been proposed that termites primarily consume the cellulose present in dung pats, which is not effectively digested by the microflora. Termites require moist dung pats and are attracted to the scent of damp, moldy, or musty wood, thriving in dark, humid environments with minimal sunlight exposure [18]. However, termites were absent, having been replaced by ants. It was posited that ants prey on both termites and Diptera larvae, leading to a reduction and eventual absence of these dipteral larvae and termites [46].

A seed was observed to germinate from a controlled dung pat, exhibiting a yellow-greenish hue. In contrast, the treated dung pat yielded fewer germinated seeds, which were characterized by a white colouration. This observation suggests that Ivermectin residues present in dung pat and soil may be toxic for dung fermenting invertebrates and might adversely affecting seed germination. Plants serve as effective bio-indicators for identifying toxic substances in the soil due to their ability to assess various parameters, including germination, growth, and genetic alterations [47]. Ivermectin has reported that high concentrations of this substance are found in plants situated near the experimental dung pats, indicating that IVM migrates from the dung pat to both the underlying soil and adjacent plants. Upon absorption by the plant, a range of biological effects can manifest [8]. Furthermore, plants may exhibit

heightened sensitivity to specific substances, serving as indicators of contamination levels in an area, thereby aiding in the preservation of biodiversity [48]. On the other hand, this experimental study placed significant emphasis on dung pat faunas, which play a crucial role in the decomposition of dung and the enhancement of soil fertility. Various groups of soil-dwelling invertebrates are subjected to AVMs, potentially resulting in biological impacts that could diminish local biodiversity [8]. Beetles are the predominant decomposers of manure in both temperate and tropical agricultural systems, with over 100 species often attracted to a single dung pile [31]. It is also a role in enhancing soil fertility through decomposition [49]. Recent studies have also identified their significance in mitigating carbon dioxide emissions from pastures [50].

Ivermectin residues affect soil fertility by the action of affecting dipteral colonization of dung and inhibiting the development of its larva. This was supported by the previous study reports of Jochmann [51] and Pecenk [52]. It was suggested that ivermectin residues can interrupt the decomposition of dung pat and lead to soil infertility. The activity of decomposers benefits humus formation in the soil and the availability of nutrients for plants. This feature is especially evident in pasture ecosystems, where an indispensable condition for correct functioning is that the cattle dung is rapidly used and transformed. When pats are not removed, they cover the pasture, inhibiting the growth of grass and a loss of available surface for cattle [53]. On the other hand, putting IVM-treated dung pat on soil land is supporting the interruption of pasture soil and the dung ecosystem. Refusal by cattle to graze on pasture grass growing near dung is a response to the offensive properties of the dung itself [52]. There was also an extreme infestation of IVM-treated dung pat by termites. Termites known by soil engineers, cause soil erosion and cause losses to seed in soil and land trees [54]. Conversely, there were reports which suggest termites transform and support soil fertility, and perform a significant contribution in upholding soil's chemical and physical parameters by excavating and breaking down organic materials when constructing their mounds [55].

## 5  Conclusion and recommendations

This experimental study demonstrates that the residual presence of Ivermectin in the environment significantly impacts dung decomposition, dung fauna, and soil fertility. The research identified dung beetles and Diptera as species affected by Ivermectin excreted through cattle feces after the withdrawal period. It was observed that Ivermectin residues in bovine dung impeded adult Diptera from visiting, ovipositing, and enabling larval survival. Moreover, dung beetles suffered considerable harm during the initial weeks following treatment. In contrast, termites, despite their affinity for dung, were not negatively impacted by Ivermectin residues. Additionally, the presence of Ivermectin in cattle dung adversely affected seed germination and plant growth. The residues inhibited the activities of insects that colonize dung. The contamination of soil with cattle dung containing Ivermectin residues raises significant concerns, potentially leading to further environmental and ecosystem disturbances. Consequently, it is essential to advise animal owners about the environmental repercussions associated with Ivermectin residues. Furthermore, to prevent Ivermectin-treated manure from coming into contact with grass and water sources.

## Acknowledgements

The authors would like to acknowledge Haramaya and Woldia University for the provision of laboratory equipment and reagent facilities. The authors are grateful to the laboratory workers. We also acknowledge the dairy owners and working staff for their willingness and commitment to support the success of this research work.

## Author contributions

**Conceptualization:** Tsedalu Yirsa.

**Data curation:** Hunde Adugna.

**Investigation:** Shiret Belete.

**Methodology:** Hunde Adugna.

**Software:** Tsedalu Yirsa, Shiret Belete.

**Supervision:** Shiret Belete.

**Visualization:** Tsedalu Yirsa.

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
