## [Decision Letter · Decision Letter 0]

2 Jan 2025

PONE-D-24-42597Experimental study on the effect of Ivermectin on cattle dung faunasPLOS ONE

Dear Dr. Yirsa,

Thank you for submitting your manuscript to PLOS ONE. After careful consideration, we feel that it has merit but does not fully meet PLOS ONE’s publication criteria as it currently stands. Therefore, we invite you to submit a revised version of the manuscript that addresses the points raised during the review process.

We look forward to receiving your revised manuscript.

Kind regards,

Rajib Chowdhury, M.Sc.; MPH

Academic Editor

PLOS ONE

2. In the online submission form, you indicated that [The data collected and used to support this article can be offered by the first or corresponding author upon request.]. All PLOS journals now require all data underlying the findings described in their manuscript to be freely available to other researchers, either 1. In a public repository, 2. Within the manuscript itself, or 3. Uploaded as supplementary information. This policy applies to all data except where public deposition would breach compliance with the protocol approved by your research ethics board. If your data cannot be made publicly available for ethical or legal reasons (e.g., public availability would compromise patient privacy), please explain your reasons on resubmission and your exemption request will be escalated for approval.

3. Please ensure that you refer to Figure 1 in your text as, if accepted, production will need this reference to link the reader to the figure.

4. Please include a caption for figure 4, 5, 6.

5. Please remove your figures from within your manuscript file, leaving only the individual TIFF/EPS image files, uploaded separately. These will be automatically included in the reviewers’ PDF.

Additional Editor Comments (if provided):

Reviewers' comments:

Reviewer's Responses to Questions

**Comments to the Author**

1. Is the manuscript technically sound, and do the data support the conclusions?

Reviewer #1: No

Reviewer #2: Yes

2. Has the statistical analysis been performed appropriately and rigorously? 

Reviewer #1: Yes

Reviewer #2: Yes

3. Have the authors made all data underlying the findings in their manuscript fully available?

Reviewer #1: No

Reviewer #2: Yes

4. Is the manuscript presented in an intelligible fashion and written in standard English?

Reviewer #1: Yes

Reviewer #2: Yes

5. Review Comments to the Author

Reviewer #1: In general terms, it is an interesting manuscript, and the message is important. It is well-written, and the figures and tables are valuable. However, I think the authors need to correct and precise many important things in order to improve the quality of the manuscript and make its acceptance possible.

Minor:

L1: add the country of the study (Ethiopia) to the title.

L22: change to “can influence negatively plant”

L26: keywords must be different from those in the title

L40, 51, 222: add a comma before the and

L48, 414: species names must be in italics. The first time that you mentioned a species you must add the author and the year of the description.

L49, 51, 448, 454: add a space between the last word and the brackets of the reference

L97, 417: add Tovar et al. (2023) Diversity

L173: add a reference to support this statement.

L220: Replace Findings with Results

Major:

- The photos in figures 1, 2, 3 and 4 must be combined on a single plate. In this way what is meant would be better understood visually and since they are in a single figure they would not be distributed throughout the document and in editorial terms complexity is saved in editing.

- Figure 5 and 6: Improve the quality of these figures, remove the background, make the bars not 3D, look for better contrast colors, and name the axes.

- It is important to add the age of the bulls and their weight, as well as the ivermectin dose given in a table.

- It is important to include in the discussion the effect of having done the experiment in the dry season where the abundance and diversity of invertebrates is much lower. How did this affect the results? If it had been done during the rainy season, would they have been different?

- The methodology they used to quantify and monitor the decomposition rate of excrement is not clear; you should detail it better in the Data Analysis section.

- L224-225: How were all these factors measured? with what instruments? with what criteria? in what units? This should be mentioned in detail in the methodology section. Review and include.

- L227: What characteristics of the germinated seeds were noted?

- Table 1: Grouping all beetles into a single category is very poor. There are many families associated with excrement with very different functions and roles such as coprophages, saprophages and predators. Putting them all in a single group does not allow us to understand their role in the process and how their interaction can be positive or negative.

- L440-441: The toxic effect of ivermectin on seeds must be studied in detail with specific experiments to evaluate this relationship.

Reviewer #2: This is relevant research that highlights the importance of pest control. However, the study makes clear the drawbacks and harms of poor pest management. It also reveals the relevance of evaluating ivermectin strategically. It is then recommended that livestock producers' supervision and advice be penetrated.

6. PLOS authors have the option to publish the peer review history of their article (what does this mean?). If published, this will include your full peer review and any attached files.

Reviewer #1: No

Reviewer #2: No

---

## [Author Response · Author response to Decision Letter 1]

13 Feb 2025

I have attached in word file as Responses to reviewers

---

## [Decision Letter · Decision Letter 1]

26 Feb 2025

Experimental study on the effect of Ivermectin on cattle dung faunas in Eastern Ethiopia

PONE-D-24-42597R1

Dear Dr. Yirsa,

We’re pleased to inform you that your manuscript has been judged scientifically suitable for publication and will be formally accepted for publication once it meets all outstanding technical requirements.

Kind regards,

Rajib Chowdhury, M.Sc.; MPH

Academic Editor

PLOS ONE

Additional Editor Comments (optional):

Reviewers' comments:

Reviewer's Responses to Questions

**Comments to the Author**

1. If the authors have adequately addressed your comments raised in a previous round of review and you feel that this manuscript is now acceptable for publication, you may indicate that here to bypass the “Comments to the Author” section, enter your conflict of interest statement in the “Confidential to Editor” section, and submit your "Accept" recommendation.

Reviewer #1: All comments have been addressed

Reviewer #2: All comments have been addressed

2. Is the manuscript technically sound, and do the data support the conclusions?

Reviewer #1: Yes

Reviewer #2: Yes

3. Has the statistical analysis been performed appropriately and rigorously? 

Reviewer #1: Yes

Reviewer #2: Yes

4. Have the authors made all data underlying the findings in their manuscript fully available?

Reviewer #1: Yes

Reviewer #2: Yes

5. Is the manuscript presented in an intelligible fashion and written in standard English?

Reviewer #1: Yes

Reviewer #2: Yes

6. Review Comments to the Author

Reviewer #1: After a detailed review of this new version, all comments have been addressed. In general terms, the manuscript improved its quality and is suitable for publication.

Reviewer #2: The publication is of utmost importance in livestock pest control, however, it is important and essential to monitor the environmental impact in areas where this technique is applied.

7. PLOS authors have the option to publish the peer review history of their article (what does this mean?). If published, this will include your full peer review and any attached files.

Reviewer #1: No

Reviewer #2: No

---

## [Editor Report · Acceptance letter]

PONE-D-24-42597R1

PLOS ONE

Dear Dr. Yirsa,

I'm pleased to inform you that your manuscript has been deemed suitable for publication in PLOS ONE. Congratulations! Your manuscript is now being handed over to our production team.

Kind regards,

on behalf of

Dr. Rajib Chowdhury

Academic Editor

PLOS ONE